# Effect of Dietary Supplementation of *Lactobacillus Casei* YYL3 and *L. Plantarum* YYL5 on Growth, Immune Response and Intestinal Microbiota in Channel Catfish

**DOI:** 10.3390/ani9121005

**Published:** 2019-11-20

**Authors:** Hongyu Zhang, Haibo Wang, Kun Hu, Liting Jiao, Mingjun Zhao, Xianle Yang, Lei Xia

**Affiliations:** 1National Pathogen Collection Center for Aquatic Animals, Shanghai Ocean University, Shanghai 210306, China; zhhy02@126.com (H.Z.); khu@shou.edu.cn (K.H.); 2Chinese Academy of Fishery Sciences, Beijing 100141, China; xuehu1110@aliyun.com (H.W.); jlttt666@163.com (L.J.); zhaomj@cafs.ac.cm (M.Z.); 3National Demonstration Center for Experimental Fisheries Science Education, Shanghai Ocean University, Shanghai 210306, China; 4Key Laboratory of Freshwater Aquatic Genetic Resources, Ministry of Agriculture, Shanghai 210306, China; 5Beijing Seasun Aquaculture BIO TECH. Co.LTD, Beijing 102488, China

**Keywords:** Channel catfish, *L. casei*, *L. plantarum*, growth, disease resistance, intestinal microbiota

## Abstract

**Simple Summary:**

Channel catfish became one of the most efficient aquaculture species due to its fast growth and considerable commercial importance, however, the intensive farming of these fish resulted in disease pressure and the eutrophic conditions. In this study, we isolated two strains of *lactobacillus*, *Lactobacillus*
*casei* YYL3 and *L. plantarum* YYL5, which can be potentially used as feed additives to promote the growth performance, disease resistance against *Edwardsiella*
*ictaluri*, and dramatically change the composition of intestinal microbiota of channel catfish.

**Abstract:**

The purpose of this study is to investigate the effect of probiotics *L. casei* YYL3 (Lc) and *L. plantarum* YYL5 (Lp) on growth performance, innate immunity, disease resistance and intestinal microbiota of channel catfish. A total of 252 catfish (67.20 ± 1.46 g) were randomly divided into 3 groups which were fed with basal diet, Lc-added (3.0 × 10^8^ cfu/g) or Lp-added (3.0 × 10^8^ cfu/g) diets, respectively. After 4 weeks of feeding, Lc significantly enhanced the growth and feed utilization of channel catfish compared with the control group (CG). Following that, the catfish were challenged with an intraperitoneal injection of 200 μL of the pathogenic *E.ictaluri* (2.0 × 10^6^ cfu/mL), the relative percent survival of Lc and Lp were 38.28% and 12.76%, respectively. High-throughput sequencing indicated Lc and Lp reduced the alpha diversity of the intestinal microbiota in channel catfish. *Lactobacillus* were overwhelming in the guts during probiotics treatment, but almost vanished away after 2 weeks post-cessation of probiotics administration. Compared to CG, Lc and Lp resulted in an increased abundance of *Pseudomonas* and decreased amount of *Aeromonas*. Functional analysis revealed that Lc treatment upregulated the relative abundance of Kyoto Encyclopedia of Genes and Genomes (KEGG) pathways including lipid metabolism, metabolism of other amino acids, metabolism of terpenoids and polyketides, xenobiotics biodegradation and metabolism, and nucleotide metabolism. Combined, our data revealed that Lc, as a feed additive at 3.0 × 10^8^ cfu/g, could promote the growth performance, disease resistance and dramatically change the composition of intestinal microbiota of channel catfish.

## 1. Introduction

Channel Catfish is one of the top farm-raised fish in the United States and China with a total production of over several billion pounds per year [1,2]. Over the years, the emergence of infectious diseases caused by bacteria becomes the primary limiting factor in intensive aquaculture production of catfish in China. Especially, the diseases enteric septicemia and motile *Aeromonas* septicemia caused by *Edwardsiella ictaluri* and *Aeromonas hydrophila* respectively are responsible for significant economic impacts to the channel catfish industry throughout the world [3,4]. Thus, broad-spectrum chemotherapeutic agents such as antibiotics has been used to treat these diseases, however, excessive and imprudent use of antibiotics leads to the emergence of resistant bacteria strains and environment and food safety problems. To mitigate these problems, probiotics have been suggested to be an alternative way for the prevention and control of fish health associated problems [5,6].

Lactic acid bacteria (LAB) have a reputation of being “generally regarded as safe” and are widely used in aquaculture [7,8]. *L. plantarum* is one of the widely used LAB in aquaculture and benefits to aquatic organisms by immunostimulation, disease resistance and growth enhancement [9,10,11,12,13]. For example, Son et al. [14] reported that treatment of a diet containing *L. plantarum* at the level of 10^8^ cfu/g for 4 weeks significantly increased percent weight gain and feed efficiency, and reduced their mortality rate more than 20% after challenged with *Streptococcu s*sp. Beck et al. [15] showed that *L. plantarum* feeding for 30 days increased survival rate of olive flounder which was challenged with *E. tarda* (10^5^ cfu/g fish) compared with control, probably through enhancing expression of proinflammatory genes (T-bet, IL-1β, and IFN-γ) and CD18. In Giri’s study [16], they found that administration of *L. plantarum* (VSG3) for 60 days significantly improved the growth, immunity and disease resistance of *Labeo rohita*. 

Compared with *L. plantarum*, less attention was paid to *L. casei*, although it is also commonly used in aquaculture [17]. Lamari et al. [18] demonstrated that *L. casei* X2 improved the growth performance of sea bass larvae. Abasali and Mohamad [19] found that *L. casei* in combination with *L. acidophilus*, *Enterococcus faecium*, and *Bifidobacterium thermophilum* could increase the gonadosomatic index and the production of fingerlings *Xiphophorus helleri* at reproductive age.

We previously isolated Lc and Lp from commercial probiotics, which showed significant inhibitory activity against the main bacterial pathogens, i.e., *E. ictaluri*, *A. hydrophila* and *Yersinia ruckeri*, in channel catfish in vitro [4,20] (data not published). However, their in vivo activities are unknown, especially, their effects on growth, immunity response, and intestinal bacteria of channel catfish remain elusive. Therefore, the present study was conducted to investigate and compare the efficacy of Lc and Lp on the growth performance, immunity, gut microflora and disease resistance against *E. ictaluri* in channel catfish.

## 2. Materials and Methods 

### 2.1. Ethics Statement

This study was approved by the Animal Care and Use Committee of the Green Fish Drug Innovation Center at the Chinese Academy of Fishery Sciences.

### 2.2. Probiotic Strains and Culture Conditions

Lp and Lc tested in this study were isolated from commercial probiotics in Beijing Seasun Aquaculture BIO TECH. Co. LTD and identified based on the phenotypic characteristics and 16 S rRNA gene sequencing. They were grown at 37 °C for 24 h in MRS broth (BaseBio, Hangzhou, China), and then spun down at 5000 × g for 5 min. The pellets were washed once, and then resuspended in sterile 0.85% NaCl solution. The number of the bacterial cells in the suspension was determined by MRS agar (BaseBio) plate count [12].

### 2.3. Experimental Diets

The basal diet was modified from the work of Li et al. [21]. The formulation and proximate analysis of the experimental diets are shown in Table 1. The basal diet was used as control, and the other two diets were prepared by supplementing Lc (3.0 × 10^8^ cfu/g) and Lp (3.0 × 10^8^ cfu/g) respectively. Powdered dietary ingredients were thoroughly mixed and blended with oil, water, and appropriate amount of *Lactobacillus* until a soft dough formed. The dough was cut into particles with appropriate size through mincer, then the particles were dried down to a moisture content less than 8% in a drying cabinet using an air blower at 37℃. The feed were prepared every week and stored in sealed plastic bags at −20 °C to maintain the viability of probiotic [22,23]. The viability of the bacterial cells was assessed by MRS agar (BaseBio) plate count. All experimental diets were formulated to be isonitrogenous and isocaloric.

### 2.4. Fish and Rearing Conditions

Channel catfish were obtained from Beijing Longchi Aquaculture Farm, Beijing, China, showing no signs of disease through gross and microscopic examination of skin, gills, and guts of representative samples. Upon arrival at laboratory, fish were stocked in four 3 m^3^ tanks and acclimatized with laboratory conditions for 2 weeks. Thereafter, 252 fish (67.20 ± 1.46 g) were randomly distributed into 9 glass aquariums (300 L) at density of 28 fish per tank and fed experimental diets for 4 weeks. During the feeding trial, fish were fed experimental diets twice per day (at 9:00 a.m. and 4:00 p.m.) up to apparent satiation. Any uneaten portion was collected after feeding and immediately dried in an oven at 80℃. The amounts of all diets fed were calculated by subtracting the uneaten portions and recorded daily. During the culture period, one-third of the water was renewed every 2 days, and the water quality was maintained at the following range: water temperature 26.0–27.0 °C, pH 7.2–7.8 and dissolved oxygen 6.2−6.7 mg/L. Fish in each aquarium were group weighted and counted at the end of the trail to determine weight gain (WG) and survival. WG, survival rate, specific growth rate (SGR) and feed conversion ratio (FCR) were calculated using formulae described by Gupta et al. [24].
(1)WG(g)=final weight−initial weight
(2)$$Survival\;rate(\%)=\left(\frac{fish\;no.\;of\;fish}{initial\;no.\;of\;fish}\right)\times 100\%$$
(3)$$SGR(\%)=\left(\frac{\ln\;final\;body\;weight-\ln\;initial\;body\;weight}{period\;of\;culture(days)}\right)\times 100\%$$
(4)$$FCR(\%)=\left(\frac{total\;dry\;feed\;fed(g)}{total\;live\;weight\;gain(g)}\right)\times 100\%$$

### 2.5. Non-Specific Immune Parameters of Serum

After 4 weeks growth trial, 15 fish from each group were euthanized on ice for analysis of non-specific immune parameters of serum. Blood samples were collected through the caudal vein from 5 fish per tank using a 1 mL syringe at the end of the feeding trial, and immediately transferred the Eppendorf tubes without anticoagulant, after blood tubes have been allowed to clot (1 h at room temperature and 4 h at 4 °C) and centrifuged (1500 × g, 5 min, 4 °C), the serum was obtained and stored at −20 °C until assay [25].

Alternative complement pathway activity (ACH_50_) was determined using the method described by Doan [25]: (5)$$ACH_{50}(units/ml)=1/K\times r\times 1/2$$

Where K is the amount of serum giving 50% hemolysis, r is the reciprocal of the serum dilution, and 1/2 is the correction factor. The assay was performed on a 1/2 scale of the original method [25,26].

Serum lysozyme (LZM) and Superoxide dismutase (SOD) activities were determined by LZM and SOD Assay Kits (Nanjing Jiancheng Bioengineering Institute, China) according to the manufacturer’s instructions.

### 2.6. Challenge Test

*E. ictaluri* was kindly provided by Dr Zhou (Yangtze River Fisheries Research Institute, Chinese Academy of Fishery Sciences). The strain was incubated at 28 °C, 150 rpm for 24 h in nutrient broth media (BaseBio). The cultures were centrifuged at 5000 × g for 5 min. The pellets were washed twice with sterile 0.85% NaCl solution. The number of the bacterial cells in the suspensions was measured by nutrient agar (BaseBio) plate count. At the end of the feeding trail, 54 fish from each group (Lc, Lp and CG) were challenged intraperitoneally with 200 µl (2.0 × 10^6^ cfu/mL) of *E. ictaluri*, and then redistributed evenly to 3 glass tanks. The challenge test lasted for 15 days, and all fish were fed twice daily with basal diet. Fish mortality for each tank was recorded daily, and cumulative mortality and RPS were calculated by the following formula [27].
(6)$$Cumulative\;mortalatiy(\%)=\left(\frac{Total\;mortality\;in\;each\;treatment\;after\;challenge}{Total\;number\;of\;fish\;challenged\;for\;same\;treatment}\right)\times 100$$
(7)$$RPS(\%)=\left(1-\frac{Percentage\;mortality\;in\;probiotic\;groups}{Percent\;mortality\;in\;control\;group}\right)\times 100$$

### 2.7. Sample Collection and DNA Extraction

The intestinal flora was sampled after 4 weeks (right after feeding trial) and 6 weeks (2 weeks extra after cessation of probiotics), respectively. Fish were fasted for 24 h before sampling. Three samples (one fish per replicate) randomly selected from each group at each time point, were anesthetized with eugenol, brain tissue was destroyed with anatomic needle. The abdomen of fish was cleaned with 70% ethanol, and the abdominal cavity was opened under aseptic condition. The whole intestine was taken out and 1 cm anterior and posterior intestine was discarded. The intestinal contents were extruded into a 1.5 mL sterile tube, then the whole intestinal tract was dissected, rinsed with sterile phosphate buffer saline (PBS) for three times, and centrifuged to collect the residual content.

Bacterial DNA was extracted using the E.Z.N.A Mag-Bind Soil DNA Kit (Omega, Norcross, GA, USA) according to the manufacturer’s protocol, and Qubit® 2.0 (life, Carlsbad, CA, USA) was used to measure the concentration of the DNA. A total of 18 DNA samples were submitted to Sangon Biotech, Inc. (Shanghai, China) for PCR amplification and Next-Generation Sequencing using Illumina Miseq platform. The V3-V4 region of the bacteria 16 S ribosomal RNA gene from each sample were amplified using the bacterial universal primer 341F (5’-CCTACACGACGCTCTTCCGATCTG(barcode)CCTACGGGNGGCWGCAG-3’) and 805R (5’-GACTGGAGTTCCTTGGCACCCGAGAATTCCAGACTACHVGGGTATCTAATCC-3’), where barcode is an six-base sequence unique to each sample. The PCR was performed as described previously [28]. AMPure XP beads was used to purify the free primers and primer dimer species in the amplicon product, and universal Illumina adaptor and index was used for library construction. Before sequencing, the DNA concentration of each PCR product was determined using a Qubit® 2.0 Green double-stranded DNA assay and it was quality controlled using a bioanalyzer (Agilent 2100, Santa Clara, CA, USA). Depending on coverage needs, all libraries can be pooled for one run. The amplicons from each reaction mixture were pooled in equimolar ratios based on their concentration. Sequencing was performed using the Illumina MiSeq system (Illumina MiSeq, San Diego, CA, USA), according to the manufacturer’s instructions.

### 2.8. Bioinformatic Analysis 

The paired end sequence reads were quality trimmed using the Cutadapt tool (Version 1.2.1) to remove adaptors, barcodes, primer sequences. Paired end reads were merged using the PEAR tool (Version 0.9.6) [29], and low-quality reads (Q score <20) were removed using Prinseq (Version 0.20.4) [30]. After de-multiplexing and quality filtering of the raw sequence reads, reference-based and de novo chimeras were checked and removed from the cleaned sequences and operational taxonomic unit (OTU) clustering was performed with a 0.97 threshold using Usearch (Version 5.2.236) [31]. The analyses of alpha diversity indexes comprising community diversity (Simpson, Shannon index and Good’s coverage) and richness (ACE and Chao-1), which were calculated using Mothur (Version 1.30.1) [32]. The beta diversity and taxon composition were analyzed using QIIME (version 1.8) for calculating weighted UniFrac. Principal coordinates analysis (PCoA) was used to evaluate the Beta diversity obtained by weighted UniFrac analyses [33]. Weighted PCoA 3D figure was created using the vegan (version 2.0-10) in R (version 3.2), and heatmap of genus was generated using gplots (version 2.17.0) package in R (version 3.2).

To predict the functional profiles of microbial communities, the Phylogenetic Investigation of Communities by Reconstruction of Unobserved States (PICRUSt) method was used [34]. Briefly, 16S rRNA gene sequences were clustered into OTUs using the closed reference OTU picking algorithm and the Greengenes reference taxonomy (Greengenes 13.5). The predicted 16S copy number was used to normalize the OTU table for each representative sequence. Molecular functions for each sample were predicted by categorizing annotated metagenome sequences using KEGG Orthology database into KEGG pathways. A two-sided Welch’s t-test was used to identify enriched metabolic pathways in the microbiota of catfish fed with probiotics by software STAMP, with *p* < 0.05 considered significant.

### 2.9. Statistical Analysis

All data are expressed as mean ± SD. The statistical significance between means of the independent groups was analyzed using one-way ANOVA followed by LSD (Least-Significant Difference), and *p* value less than 0.05 was statistically significant.

## 3. Results

### 3.1. Growth Performance

Growth parameters at the end of 4 weeks feeding trail are presented in Table 2, the Lc has significant higher WG and SGR than those in the CG and Lp. Besides, the Lc showed significant lower FCR than the CG (*p* < 0.05). No significant difference in survival rate was observed among Lc, Lp, and CG (*p* > 0.05).

### 3.2. Immune Parameters

After 4 weeks feeding trail, Lc significantly enhanced LZM activity than CG and Lp group (*p* < 0.05), but Lp has no effect on LZM compared with CG (*p* > 0.05) (Figure 1A). Lc and Lp didn’t affect ACH_50_ and SOD compared with CG (Figure 1B,C).

### 3.3. Challenge Test

The challenge test showed that supplementation of Lc enhanced the protection against *E. ictaluri* infection. Lc (53.70 ± 8.49) % had lower cumulative mortality than CG (87.02 ± 6.42) % and Lp (75.93 ± 11.56) % (*p* < 0.05) (Figure 2). The RPS of Lc and Lp were 38.28% and 12.76%, respectively. 

### 3.4. Characteristics of the High-Throughput Sequence Data

After quality filtration and adapter trimming of raw reads, a total of 1,024,202 valid sequences were obtained from all channel catfish intestinal microbiota samples. Then they are clustered into 8727 OTUs using a 97% sequence identity cutoff. To assess the sequencing depth and species richness, a rarefaction curve was constructed for each sample.

The rarefaction curves (Figure 3) indicated that the sufficient sampling depth was achieved for each sample. 

### 3.5. Diversity Analysis

To compare the bacterial diversity across groups, bacterial richness and diversity indices were calculated from the proportion of OUTs (Table 3). The Good’s coverage of each group was above 99%, indicating an adequate depth of sequencing. Right after 4 weeks probiotics feeding trials, CG had significantly higher species richness estimated by ACE (1041.25 ± 199.74) than Lc (472.49 ± 316.03) and Lp (531.92 ± 148.74), and also higher bacterial diversity estimated by Simpson index (0.21 ± 0.06) than Lc (0.75 ± 0.28) and Lp (0.73 ± 0.19). Two weeks feeding of basal diet after cessation of probiotic administration resulted in that the community diversity in Lp-2wk, but not Lc-2wk, recovered to the levels of CG and CG-2wk as evidenced by ACE, Chao1 index, and Goods coverage.

A PCoA analysis of the weighted UniFrac distances (Figure 4) displayed that the samples were segregated by the type of diet, with the Lc and Lp samples clearly distinct from the CG samples right after 4 weeks probiotic feeding trials. Following 2 weeks of basal diet feeding after cessation of probiotic tests, Lp-2wk and CG-2wk grouped together, which were separated from Lc-2wk group, suggesting that the diversity of Lp-2wk was closer to CG-2wk than Lc-2wk. This was also supported by heatmap of genus results which showed that Lp-2wk, CG and CG-2wk were clustered together (Appendix A).

### 3.6. Changes in Community Structure and Intestinal Microbiota Abundance in the Catfish

To further investigate the changes of gut microbial communities after feeding of probiotics, bacterial phyla and genus frequencies from 3 groups of samples were analyzed (Figure 5A). We found CG samples have 3 major phyla, i.e., Fusobacteria (38.25%), Bacteroidetes (29.37%), and Firmicutes (27.46%), while Lp and Lc samples were dominated by Firmicutes (accounting for >95% of 16S reads). Two weeks after cessation of probiotic administration, the intestinal flora in CG did not change much, as evidenced by the top 5 abundant phyla (Fusobacteria, Bacteroidetes, Firmicutes, Proteobacteria and Verrucomicrobia) (Appendix A). While the average number of Firmicutes in Lc-2wk and Lp-2wk dropped sharply from >95% to 6.69% and 0.53%, respectively. Their dominant phyla become Proteobacteria (81.4%) and Fusobacteria (84.72%), respectively.

Similarly, at the genus level (Figure 5B), the intestinal flora in CG changed slightly between two time points, while they altered significantly in Lc and Lp groups, e.g., *Lactobacillus* as expected was the most abundant (accounting for >80% of 16S reads) genus in both Lc and Lp groups, however, this genus almost disappeared (<1%) after two weeks post-cessation, suggesting that *Lactobacillus* is a transient bacteria strain. Intriguingly, the intestinal bacteria in both group had different changes, the expression of multiple other genera increased in Lc-2wk, such as, *Pseudomonas* (19.75%), *Ancylobacter* (13.41%), *Neorhizobium* (10.47%), *Mesorhizobium* (9.10%), *Brucella* (8.26%), *Ralstonia* (6.06%), *Pseudoclavibacter* (3.69%), *Phenylobacterium* (2.56%), *Luteimonas* (2.06%) and *Anaerobacter* (2.05%). However, the predominant genera in Lp-2wk were *Cetobacterium* (84.72%) and *Akkermansia* (12.23%).

### 3.7. Functional Analysis of the Intestinal Microflora

To further analyze the effect of probiotics on gut bacteria, we performed functional analysis based on the community composition of the gut microbiome after probiotics-fed via PICRUSt. KEGG orthologs were classified to level 3. Most of the predicted functional pathways belonged to six main categories, i.e., metabolism, organismal systems, genetic information processing, cellular processes, environmental information processing and human diseases (Figure 6). Although there were no significant different abundances among Lc, Lp and CG in metabolism (*p* > 0.05), five KEGG pathways (Lipid Metabolism, Metabolism of Other Amino Acids, Metabolism of Terpenoids and Polyketides, Xenobiotics Biodegradation and Metabolism and Nucleotide Metabolism) in level 2 and 20 KEGG pathways in level 3 were significantly enriched in Lc (*p* < 0.05). One KEGG pathway (Metabolism of Terpenoids and Polyketides) in level 2 and 10 KEGG pathways in level 3 were significantly enriched in Lp (*p* < 0.05) (Appendix A).

## 4. Discussion

LAB as probiotics have been widely used as dietary supplements for promoting growth performance and disease resistance of aquatic animals including tilapia [35], common carp [36], sea bass [18], rainbow trout [37], red sea bream [38], and olive flounder [39]. However, few probiotics have been used for channel catfish. Previously, we isolated two new LAB strains, *L. casei* YYL3 and *L. plantarum* YYL5, with potential antibacterial activity against *E. ictaluri*, *A. hydrophila*, and *Y. ruckeri*, which are common bacterial pathogens of catfishes. In the present study, we found significant increase in growth parameters (WG and SGR) and decrease in FCR in catfish fed with Lc, at 10^8^ cfu/g level for 4 weeks compared with CG. This is in congruence with the effects of other *L. casei* strains on aquatic animals. For instances, Lamari et al. [18] reported that *L. casei* X2 could promote the growth of sea bass larvae. Andani et al. [40] found that commercial feed containing 5 × 10^7^ cfu/g of *L. casei* could improve growth parameters of rainbow trout. This growth enhancement by Lc is probably due to its effect on intestinal microbiota which has been demonstrated to play important roles in digestion, the production of essential vitamins, and protection of the gastrointestinal tract from pathogen colonization [41]. In accordance with this assumption, we found that Lc-feed changed the relative abundance of many genes enriched in metabolism pathways including lipid metabolism, metabolism of other amino acids, metabolism of terpenoids and polyketides, xenobiotics biodegradation and metabolism, nucleotide metabolism and carbohydrate metabolism, which could potentially improve feed utilization and growth.

Stimulation of immune system and increase of disease resistance are another important benefits of probiotics [42]. Lysozyme plays an important role in disease defense by its antibacterial activity [43], thus its activity has been frequently used as an indicator of non-specific immune functions, which is of primary importance in combating infections in fish. In this study, Lc supplementation significantly upregulated lysozyme activity in the blood of catfish, which is coincide with higher RPS in Lc following challenge with *E. ictaluri*. This indicates that dietary supplementation of Lc can improve disease resistance in catfish in comparison with control and Lp. Similarly, significant stimulation of fish immunity by dietary Lc has been reported in previous studies on zebrafish [44], and rainbow trout [40].

Although the positive effects of *L. plantarum* in aquaculture animals have been investigated including enhancement of the growth performance [11,14,25], and immune response [15,35,45] and inhibition of adhesion of pathogenic microorganisms [9,10]. In this study, we didn’t find any significant effect of Lp on growth performance, immune parameters and RPS of catfish compared to CG. Similar to our study, Lee et al. [46] revealed that oral administration of diet with different levels of *L. plantarum* KCTC3928 (10^6^ ~ 10^8^ cfu/g) could not improve growth performance of Japanese eel in terms of weight gain, feed efficiency ratio, and protein efficiency ratio. Beck et al. [45] found that the single *L. plantarum* FGL0001-fed group of olive flounder did not produce a statistical increase in weight gain compared to the control. Butprom et al. [47] found that *L. plantarum* C014 could promote lysozyme activity in hybrid catfish after 45 days of feeding but not 30 days. The observed different effects of *L. plantarum* on growth performance and immune response of different aquatic animals may be attributed to the host species-specific response, and different active ingredients secreted by different *L. plantarum* strains.

There are several studies on the composition of intestinal microbiota in channel catfish. Larsen et al. [48] investigated the gut microbiota of channel catfish, *Micropterus salmoides*, and *Lepomis macrochirus* from the same pond using 16S rRNA pyrosequencing, and found that microbiota differed significantly between fish species in terms of bacterial species evenness. However, all gut communities shared the dominant species *Cetobacterium somerae, Plesiomonas shigelloides* and *F. mortiferum* which belong to the phylum Fusobacteria. Bledsoe et al. [49] surveyed the intestinal microbiota of channel catfish with high-throughput DNA sequencing of 16S rRNA V4 gene amplicons derived from fish at different ages, and revealed that microbial communities inhabiting the intestines of catfish early in life were dynamic, with sharp transition occurring up to 125 days post hatch (dph) when the microbiota somewhat stabilized. The most abundant microbiota in the catfish after 3 dph are Bacteroidetes, Firmicutes, Fusobacteria, and Proteobacteria, with the species *C.somerae* and *P. shigelloides*. In agreement with these findings, our study showed that the Fusobacteria, Bacteroidetes, Firmicutes and Proteobacteria were the dominant phyla, with the genus *Cetobacterium* showing the highest abundance in the control group (120dph–150dph).

After 4 weeks feeding trail, *Lactobacillus* reduced the alpha diversity of the intestinal microbiota. At the genus level, *Lactobacillus* were the dominant bacteria of Lc and Lp groups, and the proportion of *Lactobacillus* were 85.39% and 84.58%, respectively. *Pseudomonas* was the second and third most dominant microbiota of Lc and Lp groups, separately. After 2 weeks cessation of probiotic administration, the proportion of *Pseudomonas* increased from 1.59% in Lc to 19.75% in Lc-2wk, becoming the first dominant bacteria, while this bacterium was not detected in CG, Lc, Lp and CG-2wk, and only a very low level in Lp-2wk. Because this bacterium that is widely distributed in the fish and shrimp, can produce lipase, increase lipid utilization rate and improve growth performance [50,51,52], it has been regarded as candidate probiotics in grass carp and is an important biological control agent in aquaculture [53]. *Aeromonas* are thought to be opportunistic pathogens in freshwater fish, which exist in the breeding environment and intestinal tract. Bacterial diseases occur easily when the immunity of fish declined, or the environment changed. In our study, the relative abundance of *Aeromonas* in the intestinal of CG increased from 0.13% to 6.43% in CG-2wk. It is reported that fish digestive tract is a reservoir for many opportunistic pathogens [54,55], whereas both Lc and Lp exhibited significant inhibition of *Aeromonas*, suggesting that the Lc and Lp dramatically changed the composition of intestinal microbiota of channel catfish.

## 5. Conclusions

This study evaluated two probiotics Lc YYL3 and Lp YYL5 in channel catfish aquaculture. Dietary supplementation with Lc YYL3 improved the growth performance and disease resistance in channel catfish. Dietary supplementation with Lc YYL3 or Lp YYL5 reduced the alpha diversity and dramatically changed the composition of the intestinal microbiota in channel catfish.

## Figures and Tables

**Figure 1 animals-09-01005-f001:**
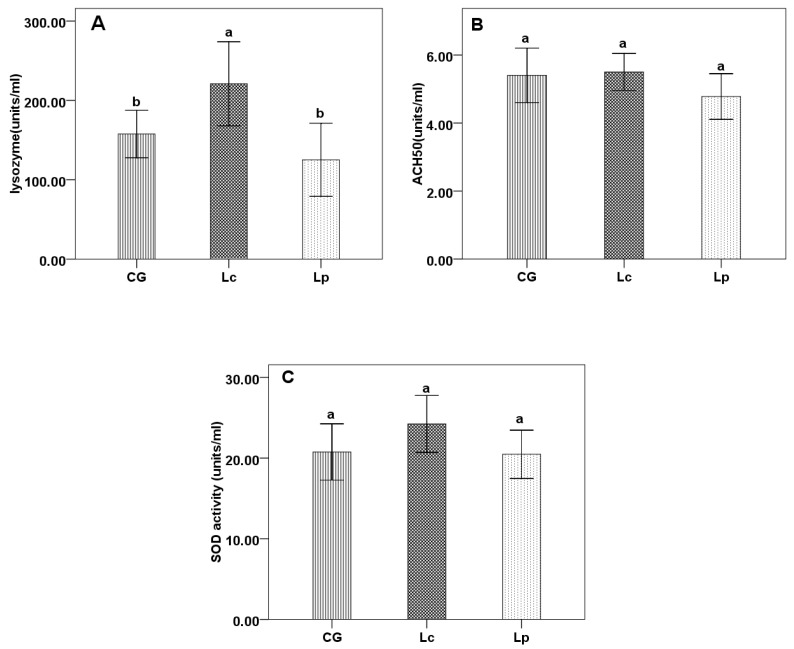
Lysozyme (**A**), alternative complement pathway activity (ACH_50_) (**B**), and superoxide dismutase (SOD) (**C**) of channel catfish fed the three diets for 4 weeks. CG: basal diet; Lc: basal diet with *L. casei* YYL3; Lp: basal diet with *L. plantanum* YYL5. Each value represents mean ± SD (n=5). Different letters are significantly (*p* < 0.05) different by LSD test.

**Figure 2 animals-09-01005-f002:**
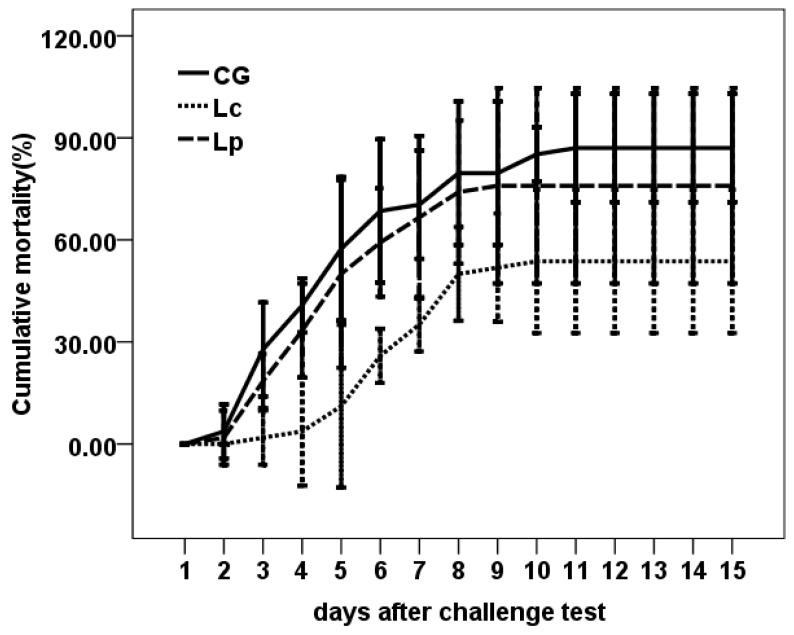
Cumulative mortality of channel catfish after challenged with the pathogen, *E. ictaluri*.

**Figure 3 animals-09-01005-f003:**
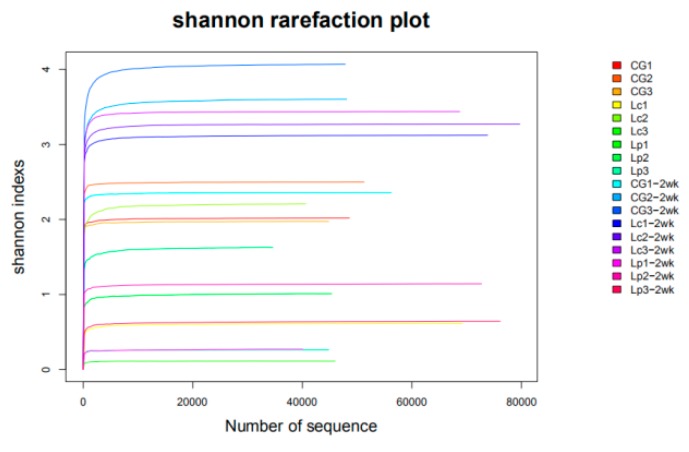
Shannon rarefaction curves of catfish intestinal microbiota samples which were harvest at 4 and 6 weeks. Note: CG: control group fed with basal diet for 4 weeks; Lc: fed with basal diet supplemented with 3.0 × 10^8^ cfu/g of *L. casei* for 4 weeks; Lp: fed with basal diet supplemented with 3.0 × 10^8^ cfu/g of *L. plantarum* for 4 weeks; CG-2wk: CG group continue to feed basal diet for two more weeks; Lc-2wk: Lc group feeding basal diet instead of the probiotics diet for two more weeks; Lp-2wk: Lp group feeding basal diet instead of the probiotics diet for two more weeks. The following is the same.

**Figure 4 animals-09-01005-f004:**
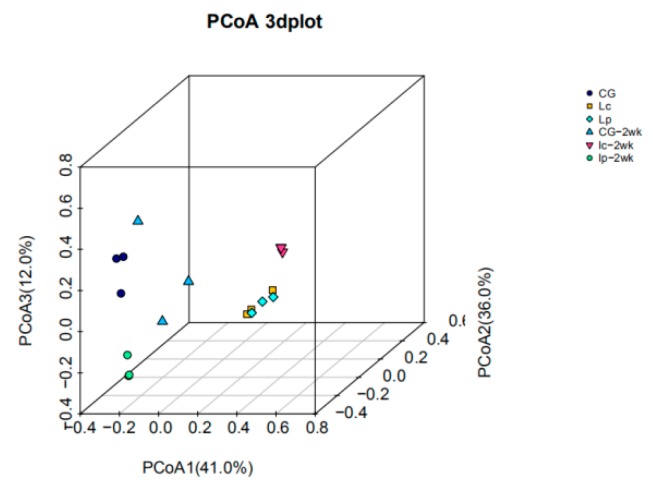
PCoA 3D figure of community compositions in catfish intestinal microbiota based on UniFrac distance matrix. CG, blue circle; Lc, yellow square; Lp, cyan diamond; CG-2wk, cyan triangle; Lc-2wk, purple triangle; Lp-2wk, green circle.

**Figure 5 animals-09-01005-f005:**
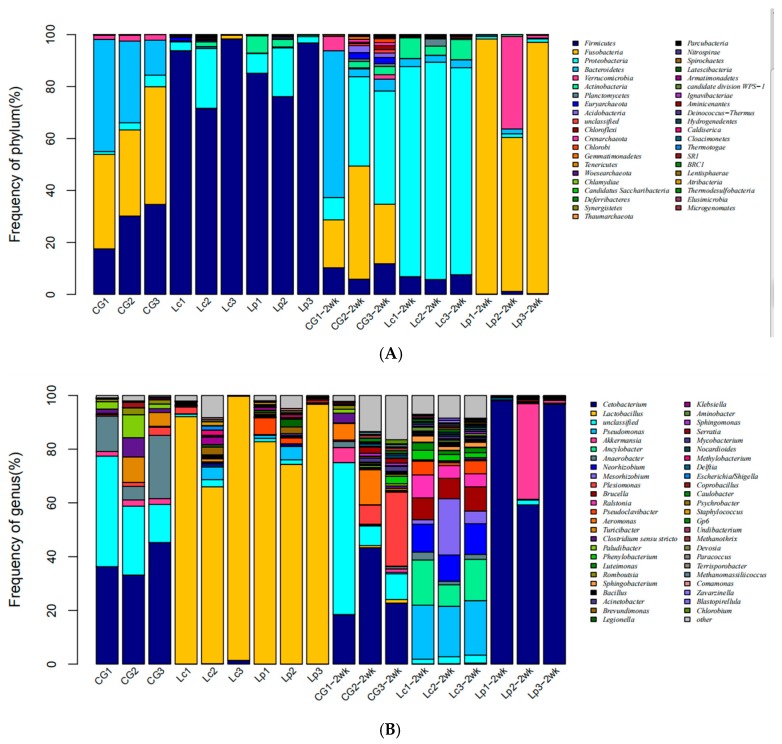
Gut bacterial community structure at phylum (**A**) and genus (**B**) levels. Each treatment has three replicates.

**Figure 6 animals-09-01005-f006:**
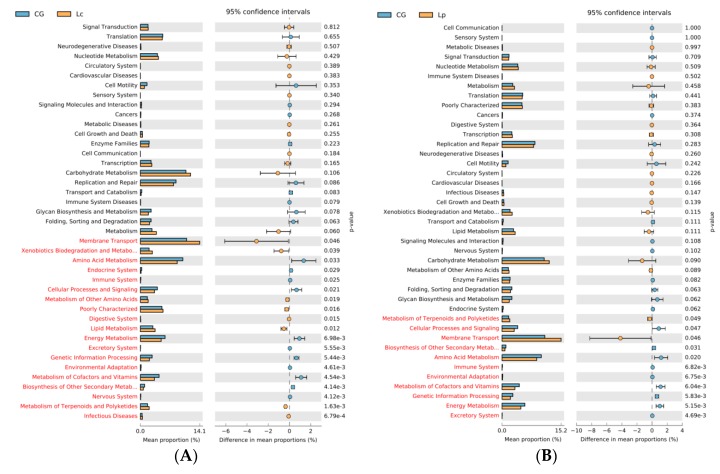
Comparison of normalized abundance data of KEGG pathways of intestinal microbiome from CG group with Lc group (**A**) and with Lp group (**B**).

**Table 1 animals-09-01005-t001:** Composition and nutrient content of basal diet (expressed as %).

Ingredient	Proportion
Fish meal	8.00
Soybean meal	45.00
Corn meal	25.00
Wheat middling	7.40
Corn oil	3.60
Dicalcium phosphate	1.00
Vitamin mix^1^	0.50
Mineral mix^2^	0.50
Cellulose	6.00
Carboxymethyl cellulose	3.00
Proximate composition	
Moisture	7.84
Protein	30.56
Lipid	5.76
Ash	6.84

^1^ Vitamin premix (mg/kg dry diet): V_A_,8;V_D3_,2;V_K_,10;V_E_,200;V_B1_,10;V_B2_,12; V_B6_, 10; calcium pantothenate,32; nicotinic acid, 80; folic acid, 2; V_B12_, 0.01; V_H_, 0.2; choline chloride, 400; V_C_, 400. ^2^ Mineral premix (mg/kg dry diet): ZnSO_4_·7H_2_O,150; FeSO_4_·7H_2_O, 40; MnSO_4_·7H_2_O,25; CuCl_2_, 3; KI, 5; CoCl_2_·6H_2_O, 0.05; Na_2_SeO_3_, 0.09.

**Table 2 animals-09-01005-t002:** Growth performance of channel catfish fed with experimental diets for 4 weeks (mean ± SD).

	CG	Lc	Lp
Initial weight (g)	67.43 ± 1.31	67.18 ± 1.42	67.01 ± 1.64
Final weight (g)	108.40 ± 9.57 ^b^	114.67 ± 10.75 ^a^	110.57 ± 10.61 ^b^
WG (g)	40.95 ± 0.94 ^b^	47.49 ± 1.62 ^a^	43.56 ± 1.77 ^b^
Survival rate (%)	98.81 ± 2.06 ^a^	98.81 ± 2.06 ^a^	100.00 ± 0.00 ^a^
SGR (%)	1.69 ± 0.03 ^b^	1.90 ± 0.05 ^a^	1.79 ± 0.06 ^b^
FCR (%)	0.98 ± 0.07 ^a^	0.82 ± 0.02 ^b^	0.89 ± 0.04 ^ab^

^a-b^ In the same line, means with different letters are significantly different (*p <* 0.05), means with the same letters are not significantly different (*p* > 0.05). CG, basal diet; Lc, basal diet with 3.0 × 10^8^ cfu/g of *L. casei* YYL3; Lp, basal diet with 3.0 × 10^8^ cfu/g of *L. plantanum* YYL5.

**Table 3 animals-09-01005-t003:** The intestinal microbiota diversity index of each group (mean ± SD).

Sample	Shannon Index	ACE	Chao1 Index	Goods Coverage	Simpson
CG	2.16 ± 0.29 ^ab^	1041.25 ± 199.74 ^a^	600.46 ± 53.19 ^ab^	0.997 ± 0.000 ^c^	0.21 ± 0.06 ^b^
Lc	0.98 ± 1.10 ^bc^	472.49 ± 316.03 ^b^	455.37 ± 324.30 ^b^	0.999 ± 0.001 ^a^	0.75 ± 0.28 ^a^
Lp	0.97 ± 0.68 ^bc^	531.92 ± 148.74 ^b^	466.16 ± 192.22 ^b^	0.998 ± 0.001 ^b^	0.73 ± 0.19 ^a^
CG-2wk	3.34 ± 0.88 ^a^	1048.15 ± 358.86 ^a^	1008.51 ± 489.45 ^a^	0.997 ± 0.000 ^bc^	0.16 ± 0.03 ^b^
Lc-2wk	3.28 ± 0.16 ^a^	513.82 ± 30.60 ^b^	518.77 ± 22.11 ^b^	0.999 ± 0.000 ^a^	0.10 ± 0.01 ^b^
Lp-2wk	0.69 ± 0.44 ^c^	808.84 ± 60.62 ^ab^	640.22 ± 126.39 ^ab^	0.997 ± 0.000 ^bc^	0.74 ± 0.25 ^a^

Note: In the same column, different superscript letters denote significant difference (*p* < 0.05), while the same letters are not significantly different (*p* > 0.05).

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
