# Peer review of "Effect of Dietary Supplementation of Lactobacillus Casei YYL3 and L. Plantarum YYL5 on Growth, Immune Response and Intestinal Microbiota in Channel Catfish"

_animals, 2019, doi:10.3390/ani9121005_

Round 1

Reviewer 1 Report

Zhang et al., Effect of dietary supplementation of Lactobacillus casei YYL3 and L. plantarumYYL5 on growth, immune response and intestinal microbiota in channel catfish In this study, the authors evaluated investigate the effect of probiotics L. casei YYL3 (Lc) and L. plantarum YYL5 (Lp) on growth performance, innate immunity, disease resistance and intestinal microbiota of channel catfish. They found that Lactobacillus were overwhelming in the guts during probiotics treatment, but almost vanished away after 2-week 34 post-cessation of probiotics administration. They revealed that Lc, as a feed additive at 3.0×108 cfu/g, could improve growth performance, enhance immune status and disease resistance, and optimize the composition of intestinal microbiota of channel catfish. The main set-up of this experiment and analytical methods used are technically sound. The claims are convinced and supported by the experimental data.

Below are additional comments that would improve the manuscript:

Line 21 "Lactobaccilus. casei YYL3" should be "Lactobaccilus casei YYL3"

Line 23 "E. itucl" should be Full name; please check  "E. itucl"? "E. ictalur"?

Line 36 "Pseudomonas" and "Aeromonas" should be Italics.

Line 42 "optimize the composition of intestinal microbiota of channel catfish", Line 23 " optimize intestinal microbiota of channel catfish", " optimize", what is " optimization"?

Line 51 "E. ictalur" should be Full name.

Line 157-158 "Three samples (one fish per replicate) randomly selected from each group at each time point" one fish per replicate, why?

Line 203-205 "A two-sided Welch's t-test was used to identify enriched metabolic pathways in the microbiota of catfish fed with probiotics by software STAMP, with P< 0.05 considered significant." three groups were determined, "t-test" used here is wrong.

Line 200-201 "The known 16S copy number was used to normalize the OTU table for each representative sequence." "known 16S copy number", where do you find the known 16S copy number? Some are unknown. If you do not know the copy number of some unknown bacteria, then how do you deal with these data? Please give a reference.

Line 248-249 In results, the authors should present concrete data, not descriptive words, such as "satisfactory".

Line 271-277 Pay attention to tense. Past or present tense?

Reviewer 2 Report

In the presented manuscript, Hongyu Zhang with colleagues aims to investigate the effect of two Lactobacillus-based probiotics on growth performance, disease resistance and the composition of the intestinal microbiota of channel catfish. The manuscript is well written and even though it presents some intriguing findings, it is very descriptive and lacks any deeper insights into the causality of the observations.

Minor issues

I would suggest a change of title, particularly because the study does not investigate any effect on immune response, but only the effect on three parameters of innate immunity, which undergo only marginal changes On a similar note in a SIMPLE SUMMARY and ABSTRACT..statements about enhanced immunity (line 22) and significant enhancement of immune parameters (line 29) do not reflect the content of the manuscript. Out of the three measured parameters of innate immunity, only one was significantly enhanced. Furthermore, with the exception of increased survival of the catfish following the challenge with E. ictaluri, the authors did not provide any demonstration that the immunity is enhanced. Please delete or reformulate these statements. For the challenge experiment, authors use IP injection of E. inctaluri. I wonder what mechanism of protection do the feeds provide. The measured immune parameters (which are only marginally changed) do not seem to explain such a significant reduction in mortality. I wonder what is the link between the functional feeds and increased survival following the IP challenge. Please explain and provide examples from the literature. Both feeds (LP and LC) induce substantial changes in the composition of the gut microbiome (including an overall decrease in the diversity of bacterial communities) which are long-lasting and the microbiota does not recover to the initial composition even two weeks post-feeding. Why do authors assume these changes are positive. Similarly, please modify the statement from the line 42 – the feed may promote the growth performance, disease resistance and dramatically change the composition of the intestinal microbiota. However, no data are provided supporting the “optimization of the composition”. If you insist on using the word “optimize” please specify how is the new feed more optimal than control? The authors investigate changes in the relative abundance of genes involved in metabolic pathways in LC and LP feed groups. Considering how dramatic are the overall changes in the microbiota composition, it is surprising to see how marginal are the changes in the proportion of these genes in KEGG pathways.  Could the authors please elaborate on this in the discussion? Throughout the manuscript, space is missing in front of the majority of the brackets.